# Forecasting the spread of SARS-CoV-2 is inherently ambiguous given the current state of virus research

**Melissa Koenen** , **Marleen Balvert**, **Ruud Brekelmans, Hein Fleuren**,
**Valentijn Stienen, Joris Wagenaar***

Zero Hunger Lab, Department of Econometrics and Operations Research, Tilburg School of Economics and
Management, Tilburg University, Tilburg, The Netherlands

* j.c.wagenaar_1@tilburguniversity.edu

## Abstract

Since the onset of the COVID-19 pandemic many researchers and health advisory institutions have focused on virus spread prediction through epidemiological models. Such models rely on virus- and disease characteristics of which most are uncertain or even unknown for SARS-CoV-2. This study addresses the validity of various assumptions using an epidemiological simulation model. The contributions of this work are twofold. First, we show that multiple scenarios all lead to realistic numbers of deaths and ICU admissions, two observable and verifiable metrics. Second, we test the sensitivity of estimates for the number of infected and immune individuals, and show that these vary strongly between scenarios. Note that the amount of variation measured in this study is merely a lower bound: epidemiological modeling contains uncertainty on more parameters than the four in this study, and including those as well would lead to an even larger set of possible scenarios. As the level of infection and immunity among the population are particularly important for policy makers, further research on virus and disease progression characteristics is essential. Until that time, epidemiological modeling studies cannot give conclusive results and should come with a careful analysis of several scenarios on virus- and disease characteristics.

## Introduction

The COVID-19 pandemic has disrupted society all across the world. At the time of the SARS-CoV-2 virus outbreak in Wuhan province, China went into lockdown. Many countries across the world followed when the virus reached them a few weeks or months later. Since then many researchers and national health institutions have focused on predicting the course of the epidemic, assessing the effects of non-medical interventions in the form of social distancing, and evaluating the possibilities of an exit strategy [1–3]. The epidemiological models underlying these studies heavily rely on virus and disease characteristics such as the case fatality ratio (CFR). Within just a few months researchers made great progress in estimating these characteristics [4–7], and a plethora of data sources and scientific studies rapidly became available

**Data Availability Statement:** The resulting
transition matrices are available on our GitHub
page https://github.com/zero-hunger-lab/covid-

paper-supplement or via the Supplementary information.

**Funding:** The author(s) received no specific funding for this work.

**Competing interests:** The authors have declared that no competing interests exist.

[8–10]. These sources however report various estimates on parameters that describe the virus behavior and disease progression. As a result, many aspects of the SARS-CoV-2 virus' behavior that forecasting models rely on, including CFR, still remain uncertain or unknown.

The aim of this paper is to get a good view on the spread of the virus and its characteristics as the virus spread would behave without taking any social distancing measures. Background is that, in future research, we are interested how the SARS-CoV-2 virus spreads in low income countries, slums and refugee camps where, for various reasons, measures hardly can be taken or are not effective at all. We therefore use data of the initial phase of the COVID-19 spread in the Netherlands where relatively plenty of good quality data is available.

This research consists of two parts. First, the validity of several assumptions on four disease and virus characteristics: the probability of developing symptoms, case fatality ratios, when people develop immunity, and the probability of virus transmission between an infected and a non-infected individual. We did so by simulating the spread of SARS-CoV-2 under a variety of assumptions on these four parameters using data from the Netherlands. Combinations of assumptions that lead to a predicted number of ICU admissions and death toll that resembled reality were considered plausible, while scenarios that lead to predicted ICU admissions and death toll that substantially differed from reality were considered unrealistic. As such, we obtain a set of realistic assumptions for the four model parameters, which provides better insight in the SARS-CoV-2 virus spread and disease progression.

Second, we assess the sensitivity of the model predictions with respect to the uncertainty in model inputs. Several combinations of assumptions yield a realistic number of estimated daily ICU admissions and deaths and were hence plausible scenarios. However, they gave different predictions in terms of unobservable yet important characteristics, namely the number of infections and the level of immunity among the population. This means that our current knowledge on virus- and disease characteristics is insufficient for an epidemiological modeling study to give conclusive answers concerning disease spread.

In order to carry out our analysis we developed a coarse-grained agent-based simulation model. This model holds the middle between a classical SEIR model and an agent-based simulation. As such, it allows for incorporating individual characteristics that are important determinants for virus spread and disease progression, while limiting the computational complexity of the model which enables us to simulate the entire population of the Netherlands. As mentioned, the model is applied to data from the Netherlands in the time period from February 27, 2020, the day that the first case in the Netherlands was identified, until March 25, 2020, when the first effects of social distancing measures became apparent in the data.

This paper is organized as follows. An overview of our simulation model is presented in the Materials and Methods section. The subsections in Materials and Methods describe the considered scenarios as well as further modeling details. The Results section first presents results for the analysis on the validity of model parameters are provided, followed by results for the sensitivity of the population's level of infection and immunity with respect to the uncertain model parameters are presented. The paper concludes with a Discussion and Conclusions section.

## Materials and methods

In the existing literature, many researchers have already pointed out that classical SEIR models and agent-based simulations need to be adapted in order to catch all the important virus characteristics of COVID-19. Although SEIR models have been commonly used to model disease spread and form the basis of many of today's COVID-19 epidemiological models [11, 12], they need to be adapted to differentiate between age groups or geographic locations [1, 13–16]. An

often used alternative is agent-based simulation [3, 17, 18], which allows for modeling at the individual level rather than aggregating over the entire population. This is important when modeling COVID-19 [19, 20], as it allows for social and travel patterns that depend on age group and location.

The simulation model that we use to validate assumptions on virus spread and disease progression holds the middle between a compartmented SEIR model and an agent-based simulation model. Traditional compartmented SEIR models divide the population into several health stages such as susceptible (healthy individuals, denoted as S), exposed (asymptomatic infected individuals who are not able to spread the virus, E), infected (symptomatic infected individuals, I) and recovered (immune or deceased individuals, R). Over time, the health condition of individuals may progress from one health stage to another. Within the population no distinction is made between individuals based on age or other personal characteristics. Virus parameters are often estimated using differential equations. While the spread of SARS-CoV-2 as well as the disease progression behave differently depending on an individual's age and location (rural or urban), splitting the population into subgroups based on these characteristics complicates the derivation of model parameters and increases the risk of nonidentifiability. In agent-based simulation one simulates the exact daily movements of each individual. As a result agent-based simulation can take many (virus-related) individual characteristics into account and can simulate the daily contacts of each individual in detail.

While agent-based models provide the level of detail required to model SARS-CoV-2 that is lacking in SEIR models, the computational complexity of agent-based modeling is prohibitive for a population of 17 million people, which is the size of the population of the Netherlands. We therefore propose an intermediate modeling form: a coarse-grained agent-based simulation model which uses the idea of health stages to simulate agents, where agents are characterized only by their age group and geographic region. The model is akin to agent-based simulations [3, 17, 18] in that distinctive individuals are simulated who commute between their region of residence and region of employment. The main difference is that we do not include social interactions at an individual level but aggregate over groups of people with the same age, region of residence and work region. This allows us to simulate on a large scale, i.e. to simulate all inhabitants of a country or state, while including individual characteristics such as age, region of residence and commute patterns that are highly relevant when modeling the spread of SARS-CoV-2 [7, 21]. Compared to an agent-based simulation our model reduces the number of assumptions one needs to make, as we aim to simulate on a country level. Furthermore, it gives us more freedom than when using the compartment model.

The disease progression as modeled in classical SEIR models, assuming that a population consists of susceptible (S), exposed (E), infectious (I) and recovered (R) individuals, is insufficient to reflect COVID-19 [4, 11]. We extended the disease progression with several disease stages (Fig 1). First, while classical SEIR models assume that infectious individuals are symptomatic and hence observable, for COVID-19 asymptomatic individuals can be infectious as well [22–25]. We therefore split (I) into two subgroups: asymptomatic (I-a) and symptomatic (I-s). Second, it is unclear whether all infections lead to immunity [11, 26]. We tested the effects of assuming some infected individuals to not develop immunity but return to the susceptible group instead (dotted lines in Fig 1). In order to explicitly model recovered patients that obtained immunity, recovered (R) was replaced by two states: immune (IM) and deceased (D). Third, since we tested the validity of our model outcomes based on among others the number of daily ICU admissions, we included a stage "ICU admission" (ICU-a) [3, 11]. In the Netherlands, only patients with severe symptoms who have a chance of survival are admitted to the ICU [27]. For patients with severe symptoms but a low chance of survival we used

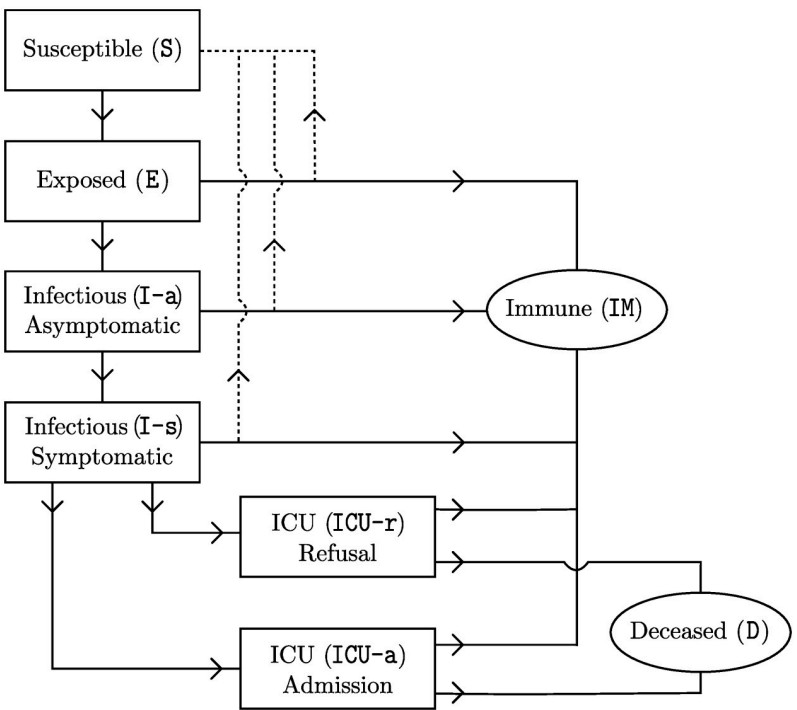

**Fig 1. Progression of disease stages in our simulation model.**

the stage "ICU refusal" (`ICU-r`). Finally, we used the classical compartments susceptible (`S`) and exposed (`E`).

The remainder of this section is organized as follows. First an overview of the tested assumptions is given, followed by the commute and contact patterns used in this study. Next the computation of the transmission probability, i.e. the probability that a non-infected individual gets infected when they meet an infectious individual, is explained. After that the health stages that an infected agent goes through are discussed. This section ends with an explanation of the initialization of the simulation model.

## Scenarios

The aim of this study was to test the validity of four important disease characteristics and assumptions. First, we estimated the probability of developing symptoms after infection. A major fraction of infections is asymptomatic and often goes by unnoticed, thus this probability is largely unknown [22]. For this, we have tested four scenarios where the probability of developing symptoms is 0.375, 0.5, 0.625 or 0.75, representing a wide range of possibilities.

Second, we assessed three scenarios for case fatality ratios. In the first scenario, we estimated the probability that a symptomatic individual dies, $P(\mathtt{D}|\mathtt{I-s})$, as the ratio between the death toll and the number of symptomatic individuals as estimated for the Netherlands. We used the death toll reported by the Dutch National Institute for Public Health and Environment (RIVM) [28] combined with the excess death rates reported by the Dutch Statistics Bureau [29]. The number of symptomatic individuals was estimated using data reported by Sanquin, the Dutch blood bank, that tested donated blood for antibodies to estimate the fraction of the population that had been infected [30]. For details see the S1 Appendix in S1 File. The second and third scenario were based on case fatality ratios per age group obtained from a study in China with approximately 72,000 cases [31]. The case counts in this study may either

include all infections, or only symptomatic cases since these are observable. Therefore, in the second scenario, we assumed that this CFR reflects the death rate among all infected individuals, $P(\text{D}|\text{E})$, and in the third scenario we assumed that CFR reflects the fatality ratio among all symptomatic individuals, $P(\text{D}|\text{I-s})$.

Third, it is yet unknown which fraction of the non-lethal infections leads to immunity. For this we considered four scenarios. In the first scenario, all infections lead to immunity. The second scenario assumed that only those individuals who develop symptoms (I-s) become immune, while others return to the group of susceptible individuals (S). In the third and fourth scenario, we assumed that having symptoms leads to immunity in only 50% and 25% of the cases, respectively, while the remaining individuals return to the susceptible group. The latter three scenarios are indicated by the dotted lines in Fig 1.

Fourth, the virus transmission probability of an infectious individual encountering a susceptible individual is unknown. In total nine different probabilities were evaluated: 0.15, 0.20, 0.25, 0.30, 0.35, 0.40, 0.45, 0.50 and 0.55. Combined with the 48 scenarios for symptom development, CFR and developing immunity, this leads to 432 scenarios in total (see Table 1).

We used data from the Netherlands to simulate the period between February 27, 2020, when the first case was identified, and March 25, 2020, when the effects of the lockdown became visible in the number of ICU admissions and the death toll. Using only the initial period of the outbreak gives the most accurate view of virus parameters (thus excluding the effect of protection measures).

## Daily commute and contact patterns

The Netherlands is divided into 40 regions termed corops, following a statistical division of the Netherlands for research institutions to present their data [32]. All approximately 17 million inhabitants of the Netherlands, termed "agents" in the simulation, have a known corop of residency and corop of employment. The population per corop per age group [33] and commute data between corop regions [34] were obtained from Statistics Netherlands. As the Netherlands is a small and densely populated country with many commuters, these corops are vastly

**Table 1. Overview of tested scenarios.**

| Input parameter | Scenario | Description |
|---|---|---|
| Probability of | 0.375 | 37.5% go from E to I-s |
| developing symptoms | 0.5 | 50% go from E to I-s |
| | 0.625 | 62.5% go from E to I-s |
| | 0.75 | 75% go from E to I-s |
| Case fatality ratio | Data from NL | $P(\text{D}|\text{E})$ based on case counts and death toll |
| | | in the Netherlands |
| | Literature-E | $P(\text{D}|\text{E})$ based on [31] |
| | Literature-I-s | $P(\text{D}|\text{I-s})$ based on [31] |
| Developing immunity | All | All infections lead to immune |
| | High | All symptomatic infections lead to immune |
| | Medium | 50% of symptomatic infections lead to immune |
| | Low | 25% of symptomatic infections lead to immune |
| Virus transmission probability | | {0.15, 0.20, 0.25, 0.30, 0.35, 0.40, 0.45, 0.50, 0.55} |

All 432 combinations are tested. E = Exposed, I-s = Infectious Symptomatic. Note that for the probability of developing immunity, we only considered infections that do not lead to death.

**Table 2. Daily contact data obtained from [35].**

| Age group | 0-9 | 10-19 | 20-29 | 30-39 | 40-49 | 50-59 | 60-69 | 70-79 | 80-150 |
|---|---|---|---|---|---|---|---|---|---|
| # of daily contacts | 12.5 | 16.1 | 21.2 | 21.8 | 22.1 | 20.9 | 15.4 | 10 | 9.5 |

interconnected. We assumed that agents who are unemployed stay in their corop of residency during the day.

The simulation divided each day into two epochs: during the day epoch, most inhabitants are at their work corop, and during the night epoch all agents are in their corop of residency. Each epoch an agent may meet other agents that reside in the same corop during that epoch, potentially leading to a new SARS-CoV-2 infection.

The daily contact pattern of an agent is determined by their age group and was obtained from [35] (Table 2). This paper reports the total number of daily contacts an individual of a certain age group has, whereas the age distribution of the people someone has contact with differs per age group. Consequently, the total number of contacts had to be divided over the different age groups. For this we used the percentage of contacts each age group has with another age group obtained from [36], by converting the amount of contacts found in that study into percentages. Combining these percentages with the total number of contacts per age group gave the contact pattern for each age group as shown in Table 3.

## Transmission probability

At every time epoch the model determined for each susceptible agent whether they got infected based on their location and age group. The probability to get infected depends on the location of the agent, the number of (infected) other agents present in that location, the contact pattern of the agent, and the probability of transmission in case the agents interacts with an infectious agent.

In mathematical terms, we defined the infection probability $p_{a,c,\,t}$ as the probability that a susceptible agent from age group $a \in A$ gets infected while being in corop $c \in C$ at epoch $t = 1$, $\cdots$, $T$. This infection probability can be determined as follows:

$$p_{a,c,t} = 1 - \prod_{a' \in A}(1 - p_{a,a',c,t})^{[\#\text{EpochContacts}]_{a,a'}},$$

**Table 3. Social patterns obtained by combining [35] and [36].**

| Age group | 0-9 | 10-19 | 20-29 | 30-39 | 40-49 | 50-59 | 60-69 | 70-79 | 80-150 |
|---|---|---|---|---|---|---|---|---|---|
| 0-9 | 4.75 | 2.25 | 1.36 | 1.36 | 0.70 | 0.70 | 0.45 | 0.45 | 0.45 |
| 10-19 | 1.93 | 6.04 | 1.87 | 1.87 | 1.34 | 1.34 | 0.56 | 0.56 | 0.56 |
| 20-29 | 1.97 | 1.99 | 2.90 | 4.34 | 2.84 | 2.84 | 1.44 | 1.44 | 1.44 |
| 30-39 | 2.03 | 2.05 | 2.99 | 4.40 | 2.92 | 2.92 | 1.48 | 1.48 | 1.48 |
| 40-49 | 1.02 | 1.33 | 2.41 | 3.16 | 3.93 | 3.93 | 2.12 | 2.12 | 2.12 |
| 50-59 | 0.96 | 1.25 | 2.28 | 2.99 | 3.72 | 3.72 | 2.01 | 2.01 | 2.01 |
| 60-69 | 0.60 | 0.60 | 0.79 | 1.32 | 1.74 | 1.74 | 2.86 | 2.86 | 2.86 |
| 70-79 | 0.39 | 0.39 | 0.51 | 0.86 | 1.13 | 1.13 | 1.86 | 1.86 | 1.86 |
| 80-150 | 0.37 | 0.37 | 0.48 | 0.817 | 1.07 | 1.07 | 1.77 | 1.77 | 1.77 |

The table shows the daily number of contacts an individual whose age group is found on one of the rows has with people from an age group found on one of the columns. For example, someone in the age group 30-39 has on average contact with 2.99 people from the age group 20-29 per day.

where [EpochContacts]$_{a,a'}$ is the number of contacts an agent of age group $a$ has with agents of age group $a'$ during an epoch (see Section "Daily commute and contact patterns") and $p_{a,a',c,t}$ represents the probability that a susceptible individual from age group $a \in \mathcal{A}$ gets infected through an encounter with an agent from age group $a' \in \mathcal{A}$, in corop $c \in \mathcal{C}$, at epoch $t = 1, \ldots, T$.

The definition of the infection probability was chosen such that it allows for region dependent probabilities as well as social patterns that depend on age groups. This makes the model flexible and realistic: suppose that in a specific corop many agents of a given age group are infected, then an agent who resides in that specific corop and meets many individuals of that age group has a high risk of getting infected.

The probability that a susceptible individual from age group $a$ gets infected through an encounter with an individual of age group $a'$, in corop $c$ at epoch $t$ ($p_{a,a',c,t}$) consisted of three components:

$$p_{a,a',c,t} = \mathbb{P}\{E_{aa'}\} \; \mathbb{P}\{I_{a',c,t}\} \; \mathbb{P}\{T\},$$

We assumed independence between these components.

- $\mathbb{P}\{E_{aa'}\}$ represented the probability that an individual from age group $a$ encounters an individual from age group $a'$. See Section "Daily commute and contact patterns" on how to obtain this probability from the contact patterns.

- $\mathbb{P}\{I_{a',c,t}\}$ described the fraction of individuals in age group $a'$ in corop $c$ that was contagious at time $t$. This fraction was determined for each epoch separately, and was computed by dividing the number of contagious agents in age group $a'$ in corop $c$ at time $t$ by the total number of agents of age group $a'$ in corop $c$ at time $t$. For the infectious agents we only included those that were in health stages I-a or I-s, as individuals with symptoms so severe that ICU admission is necessary (either ICU-a or ICU-r) were assumed to be too severely ill to (be allowed to) meet others.

- $\mathbb{P}\{T\}$ denoted the probability of virus transmission when a susceptible individual encounters an infectious individual. Since this is an unknown parameter, we tested several values for $\mathbb{P}\{T\}$, as shown in Table 1.

## Disease progression

The progression of an exposed agent's health stage from one epoch to the next was assumed to depend only on the agent's current health stage and his or her age group. Hence, after being infected, the agent's health stage over time can be interpreted as a discrete time Markov chain where transitions can occur according to Fig 1. This Markov chain is represented by a transition matrix containing the transition probabilities of an agent moving from one health stage in a certain epoch to another health stage in the next epoch. Well-known results from Markov chain analysis [37] were used to compute these probabilities such that certain pre-imposed properties were satisfied.

Two types of properties were used to compute the transition matrices. First, we used the probability that an agent will eventually reach stage $j$ at some point in time, given that the agent's current state is $i$. For example, for $i = $ E and $j = $ D, $P($D$|$E$)$ is the probability that an exposed agent will eventually decease. This corresponds to the CFR, for which we tested three scenarios as discussed in Section "Scenarios". We also used assumptions on $P($D$|$ICU-a$)$ and $P($ICU-a$|$I-s$)$. Data regarding ICU admissions and number of deaths at the ICU [38] was used to construct $P($D$|$ICU-a$)$. Furthermore, data from Sanquin on the percentage of their blood donors that had COVID-19 antibodies [30] (see S1 Appendix in S1 File) was used to

**Table 4. Age-dependent probability properties common to all scenarios.**

| Age group | $P(\text{D}|\text{ICU-a})$ | $P(\text{ICU-a}|\text{I-s})$ |
|---|---|---|
| 0–9 | $0.32 \times 10^{-2}$ | 0.00 |
| 10–19 | $2.88 \times 10^{-2}$ | $0.66 \times 10^{-4}$ |
| 20–29 | $7.99 \times 10^{-2}$ | $1.47 \times 10^{-4}$ |
| 30–39 | $15.67 \times 10^{-2}$ | $4.34 \times 10^{-4}$ |
| 40–49 | $25.90 \times 10^{-2}$ | $11.22 \times 10^{-4}$ |
| 50–59 | $38.60 \times 10^{-2}$ | $26.85 \times 10^{-4}$ |
| 60–69 | $54.03 \times 10^{-2}$ | $77.03 \times 10^{-4}$ |
| 70–79 | $71.93 \times 10^{-2}$ | $187.42 \times 10^{-4}$ |
| 80+ | $92.39 \times 10^{-2}$ | 41.86 |

obtain an estimation of the number of infections in each age group. Combining the Sanquin data with the number of ICU admissions we estimated $P(\text{ICU-a}|\text{I-s})$. The resulting probabilities that were used for estimating the transition matrices are listed in Table 4.

The second property type considered was duration, i.e. the expected time an agent spends in a health stage or a collection of health stages, given the current health stage. The values used are listed in Table 5. For instance, a well-known quantity from literature is the incubation time, i.e., the time from infection until developing symptoms. This incubation time corresponds to the expected time spent in health stages E and I-a combined, given that an agent's current health stage is E (represented in the table by $E(\text{E} \rightarrow \text{I-s})$). The average incubation time was estimated using an aggregated dataset of clinical outcomes [9] by taking the average of the reported values from the 20 studies with the largest population sizes. The average duration from symptoms to ICU admission ($E(\text{I-s} \rightarrow \text{ICU-a})$, $E(\text{I-s} \rightarrow \text{ICU-r})$) was estimated based on an average from multiple papers in literature, among which [9] and [39].

Recall that we distinguished between age groups, as many properties of COVID-19, such as the probabilities mentioned above, depend on the patient's age. Therefore, a transition matrix was estimated for each age group. The Markov properties can be used to fit a transition matrix to empirical values by minimizing a measure of goodness of fit on these properties. We used the sum of squared errors for this purpose after weighting the probability properties by a factor 100 to ensure balanced scaling between probability and duration properties.

The properties combined with the transition structure from Fig 1 do not uniquely determine the transition matrix. We therefore made the additional assumption that apart from the

**Table 5. Average duration properties of Markov chain used to fit the transition matrix.**

| Stage(s) | Description | Literature | Value |
|---|---|---|---|
| $E(\text{E} \rightarrow \text{I-s})$ | Average incubation time | [9] | 5.758 |
| $E(\text{I-a} \rightarrow \text{I-s})$ | Average duration asymptomatic infectious | - | 1.5 |
| $E(\text{I-s} \rightarrow \text{ICU-a})$ and | Average duration from symptoms until | | |
| $E(\text{I-s} \rightarrow \text{ICU-r})$ | ICU need | [9, 39] | 7.166 |
| $E(\text{ICU-a} \rightarrow \text{IM})$ and | Average time in ICU care | | |
| $E(\text{ICU-a} \rightarrow \text{D})$ | | [38] | 11.3 |
| $E(\text{ICU-r} \rightarrow \text{D})$ and | Average time in ICU-r | | |
| $E(\text{ICU-r} \rightarrow \text{IM})$ | | - | 5 |

target properties, the transition matrices of the age groups should be as similar as possible. So, rather than fitting the transition matrices for each age group independently, we fitted all transition matrices simultaneously, and extended the goodness-of-fit measure with a penalty on the pair-wise distances between the transition matrices of all age groups. The distance between two transition matrices was measured by the Euclidean norm between the two vectors containing the nonzero probabilities in these transition matrices. The resulting transition matrices are available on our GitHub page https://github.com/zero-hunger-lab/covid-paper-supplement or via the supplementary information.

## Model initialization

We estimated the number of people in each health stage for each corop on the starting date of the simulation, February 27, 2020, following the rationale explained by [40]. The duration from infection till death was estimated to be on average 24 days, based on the estimated incubation time, the time from symptoms to ICU admission, and the time from ICU until death. The number of people that got infected on day $t$ can be estimated as the number of deaths on day $t + 24$ divided by the CFR. This means that a different start situation was constructed for all CFR scenarios. To construct the start situation at February 27, day $s$, we used the daily number of deaths from March 6, the day of the first reported death, up to and including March 20, which is 24 days after the start date.

The CFR for people under 60 is very low ($<0.5\%$) leading to few deaths in these age groups. We therefore only estimated the number of infected people in the age groups 60-70, 70-80 and 80+. Assuming that the fraction of the population that got infected was the same over all age groups we estimated the number of infections under 60 per day by multiplying this fraction with the population size.

People who got infected before our simulation started, thus on day $t \in \{s - 24, s - 23, \cdots, s - 1\}$, may have progressed to other stages. We therefore determined in which stage the people who got infected on day $t \in \{s - 24, s - 23, \cdots, s - 1\}$ are at day $s$ by applying the transition matrix to the infected group for all epochs between $t$ and $s$.

The total number of infected individuals was spread over the corops following the distribution of deaths over the country as reported in [30]. All initializations constructed using this approach can be obtained from our GitHub page.

The death toll was crucial for computing the initial health stage distribution of the population. Since the actual death toll is likely higher than the number of reported COVID-19 induced deaths, we computed a lower- and an upper bound on the death toll. Two initial distributions of the population over the health stages were computed, one based on the lower bound, and one based on the upper bound. We used the average of the two as the starting point for our simulation.

The death toll reported by the National Institute for Public Health and the Environment [20] was used as a lower bound. To obtain an upper bound, we used the excess number of deaths in 2020 compared to 2015-2019. This was determined using the weekly number of deaths in the Netherlands as reported by Statistics Netherlands [29]. The excess number of deaths was computed as the number of deaths in a certain week in 2020 minus the average number of deaths in that same week for 2015-2019. The resulting excess number of deaths is 442 for week 12 and 1164 for week 13. The deaths were then distributed over the days in that week following the trend of the reported COVID-19 deaths obtained from [28]. For example, if 10% of the COVID-19 confirmed deaths in week 12 occurred on Monday, then 10% of the 442 excess deaths were added to that number. The resulting daily number of deaths are available on our GitHub page.

## Results

### Assessing the likelihood of scenarios based on number of deaths and ICU admissions

To assess the validity of each of the scenarios, we first compared the death toll of COVID-19 predicted by the simulation model with an estimate of the actual death toll in the Netherlands. The predicted death toll by the simulation is computed as an average over ten different runs to avoid outliers as the simulation has stochastic components. As the true number of COVID-19 induced deaths is highly uncertain (see e.g. [22]), we employed a reliable but safe lower and upper bound of 602 and 2139 (see S2 Appendix in S1 File for details), respectively. All scenarios that led to a prediction of the total number of deaths between February 27 and March 25 within this range were considered to be plausible. In this way, we limited the set of realistic parameter combinations to 121. Fig 2 presents a heatmap showing the deviation from the lower and upper bound on the number of deaths for all parameter combinations. A value of 0 means that the number of deaths in the simulation for the specific

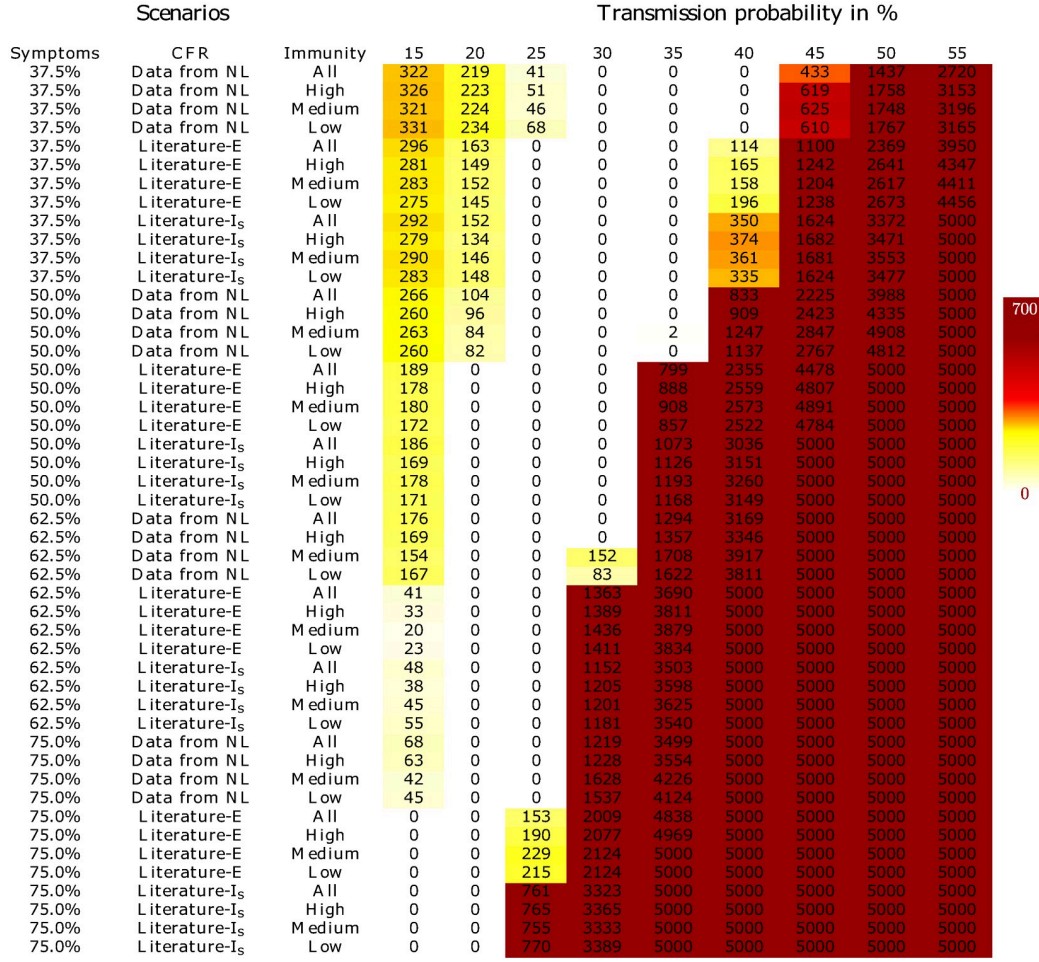

**Fig 2. A heatmap showing by how much the simulated number of deaths up to and including March 25 lies outside the interval [602, 2139] along with its legend on the right.** Columns correspond to virus transmission probabilities, rows represent the various combinations of the probability of developing symptoms, the case fatality ratio and the possibility of developing immunity. "Literature-E" and "Literature-I-s" correspond to the scenarios where the $P(D|E)$ and $P(D|I\text{-}s)$, respectively, are based on CFR estimates from the literature.

parameter combination was within the lower and upper bound and were thus acceptable parameter combinations. S3 Fig in S1 File shows the simulated total number of deaths for each of the parameter scenarios.

We further assessed the validity of the set of realistic scenarios based on the daily number of ICU admissions. Since this was a highly reliable parameter, we were able to use the reported values to compute the mean squared error (MSE) between the simulated and the real daily ICU admissions in the Netherlands. Any scenario that yielded an MSE greater than or equal to 225 was considered unreliable. The threshold of 225 was chosen based on visual inspection of plots showing simulated and real daily ICU admissions, which are available at https://covid-results.herokuapp.com. The plots allow for visual comparison of the simulations with each other and with reality, as well as a visual evaluation of the progression of these metrics over time. Furthermore, an MSE of 225 corresponds roughly to a difference between simulated and real daily ICU admissions of 15 on average. Fig 3 shows a heatmap of the MSE for only those

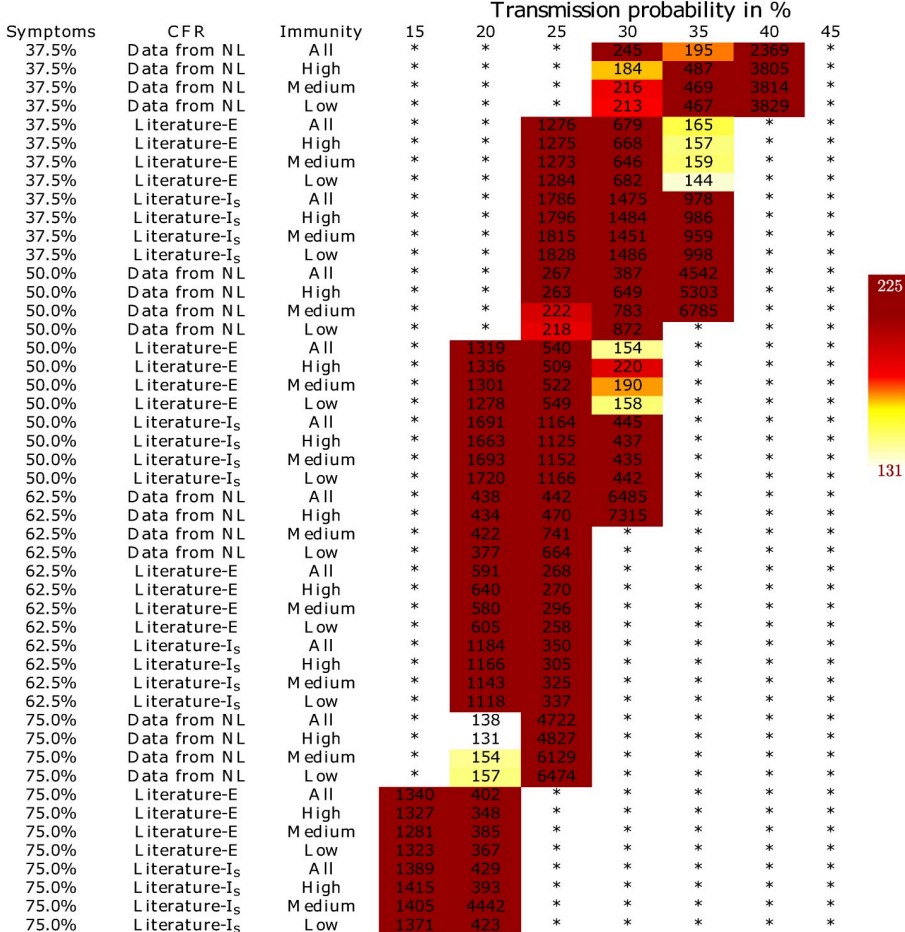

**Fig 3. A heatmap representing the prediction quality of different combinations of COVID-19 characteristics with respect to ICU occupation.** Quality is measured as the MSE between simulated and real daily ICU admissions up to and including March 25. A * indicates that the combination is not accurate in predicting the number of deaths. On the right a legend indicating that the darker the color is the higher the MSE. Columns correspond to virus transmission probabilities, rows represent the various combinations of the probability of developing symptoms, the case fatality ratio and the possibility of developing immunity. "Literature-E" and "Literature-I-s" correspond to the scenarios where the $P(D|E)$ and $P(D|I\text{-}s)$, respectively, are based on CFR estimates from the literature.

scenarios that were considered realistic based on the predicted death toll. MSEs for all scenarios are shown in S4 Fig in S1 File.

After removing all scenarios with an MSE of more than 225, our simulation still indicated 18 combinations of virus- and disease characteristics to be plausible. Note that the MSE for the ICU admission differed among these 18 settings.

## The selected scenarios lead to a wide range of predictions on the number of infected and immune individuals

For the 18 remaining parameter combinations, the left colored panel of Fig 4 shows the percentage of infected individuals up to March 12 according to our simulation. At that time social distancing measures were installed in the Netherlands, leading to a strong reduction in virus spread. Since we were interested in virus behavior under stable circumstances, i.e., without changes in human behavior, we only considered the number of infected individuals until March 12. Note that the number of infected individuals was affected by social distancing immediately, while the ICU admissions and deaths were affected only after some time, which was why we used a different time horizon for the infections (until 12 March) than with ICU admissions and deaths (until 25 March). There was substantial variation among the 18 scenarios: the percentage of the population that was infected varies from 2.0% up to 5.0%.

An interesting metric for many policy makers is the fraction of the population that develops immunity. As before, we excluded the effects of social distancing by considering infections that happened no later than March 12. Only for the individuals who were infected no later than March 12, we simulated the disease progression until they ended up in one of the stages susceptible, immune or deceased. The right colored panel of Fig 4 displays the fraction of the population that develops immunity based on the infections up to March 12. The results show major differences among the 18 scenarios: the percentage of the population that was immune varies between 0.2% and 5.2%.

| Symptoms | CFR | Immunity | Transmission Probability | % infected | % immune |
|---|---|---|---|---|---|
| 37.5% | Data from NL | All | 35% | 5 | 5.2 |
| 37.5% | Data from NL | High | 30% | 3.3 | 1.3 |
| 37.5% | Data from NL | Medium | 30% | 3.3 | 0.6 |
| 37.5% | Data from NL | Low | 30% | 3.3 | 0.3 |
| 37.5% | Literature-E | All | 30% | 2.1 | 2.3 |
| 37.5% | Literature-E | High | 30% | 2.2 | 0.9 |
| 37.5% | Literature-E | Medium | 30% | 2.1 | 0.4 |
| 37.5% | Literature-E | Low | 30% | 2.2 | 0.2 |
| 50.0% | Data from NL | Medium | 25% | 2.4 | 0.6 |
| 50.0% | Data from NL | Low | 25% | 2.4 | 0.3 |
| 50.0% | Literature-E | All | 30% | 3 | 3 |
| 50.0% | Literature-E | High | 30% | 3 | 1.5 |
| 50.0% | Literature-E | Medium | 30% | 3 | 1.5 |
| 50.0% | Literature-E | Low | 30% | 3 | 1.5 |
| 75.0% | Data from NL | All | 20% | 2 | 2 |
| 75.0% | Data from NL | High | 20% | 2 | 1.5 |
| 75.0% | Data from NL | Medium | 20% | 2.1 | 0.8 |
| 75.0% | Data from NL | Low | 20% | 2.1 | 0.4 |

**Fig 4. The percentage of infected people on March 12 (penultimate column) and the percentage of immunity among the population based on individuals who were infected no later than March 12 (last column).** The legend right next to the table indicates that the darker the color, the more people are infected or immune. Results are shown only for scenarios that lead to reasonable predictions of the death toll and the daily ICU admissions.

The percentages of infected individuals up to March 12 and percentages of immune individuals for all parameter combinations are shown in S5 and S6 Figs in S1 File.

## Discussion and conclusions

Using an epidemiological simulation model, we evaluated the likelihood of a variety of scenarios for virus spread and disease progression characteristics for the SARS-CoV-2 virus. With the four uncertain input parameters, we were able to identify 18 sets of parameter values that all led to accurately simulated daily ICU admissions and number of deaths. In particular, based on our scenarios, our analysis indicated the following conclusions (Figs 2 and 3):

1. All Literature-I-s scenarios, where we determined $P(\text{D}|\text{I}-\text{s})$ based on [31], do not seem realistic, as none of the corresponding scenarios yielded a realistic death toll and number of daily ICU admissions. Both Literature-E and Data from NL were realistic case fatality ratio scenarios, and further research is required.

2. In order for the simulation results to match reality, a high probability of symptom development required the transmission probability to be low and vice versa. According to our analysis, a low transmission probability combined with a high probability of developing symptoms was equally likely as a high transmission probability combined with a low probability of developing symptoms.

3. No conclusions regarding the probability of developing immunity could be drawn from this simulation study: the death toll and the number of daily ICU admissions differed only slightly between scenarios with different assumptions on immunity, ceteris paribus. Further research is required to obtain better estimates on the probability of developing symptoms.

While our study provided some clear pointers towards assumptions that were likely to reflect actual virus behavior, other important questions remain unanswered. This has several implications. First and foremost, modeling the spread of SARS-CoV-2 does not give conclusive insights in the number of infections and the level of immunity among the population, and requires a thorough analysis of the results for several uncertain scenarios regarding virus- and disease characteristics.

Second, the probability of developing symptoms was highly uncertain, which has major implications for virus spread predictions. When the probability for symptom development was assumed to be low, our simulation model could only reach a realistic number of ICU admissions and death toll if the uncertain parameter reflecting the transmission probability was high. This would imply a high attack rate (the percentage of the population that contracts the disease) during the early phase of the pandemic, leading to a large fraction of the community to be infected and possibly have developed immunity. On the other hand, if the probability of symptom development is high and the transmission probability is low, this would mean that only few infections have taken place so far and pre-symptomatic infections are less likely. It is thus very important to gain more insight in the probability of developing COVID-19 symptoms after infection.

Third, the progression of the virus spread in the long run was difficult to predict. If many people have already been infected, and if many infections have led to immunity, the level of immunity among the population has grown rapidly. This means that after a relatively short amount of time the susceptible group will decline and the death toll and ICU burden will reduce rapidly. On the other hand, a low number of infections and a low probability of developing immunity holds the potential of many more infections and hence deaths to come. This

is crucial for policy makers when choosing e.g. the proper level of social distancing measures and scaling up the ICU capacity.

Fourth, the haziness around immunity has major implications for policy makers. It is unclear when an infection leads to immunity and whether immunity is obtained for life [41]. Some people may even have a good immune response already at the first infection and can therefore be considered immune prior to infection. Immunity is of vital importance for political decision making when developing a vaccination policy or aiming for herd immunity. Hence, developing exit strategies is not possible without further research on how immunity works regarding SARS-CoV-2.

Epidemiological models such as ours are based on many input parameters, most of which are uncertain and only have crude estimates. In our simulation study, we already considered many scenarios by varying only four input parameters: the probability of developing symptoms, the case fatality ratio, the probability of developing immunity, and the virus transmission probability. The best available estimates from literature were used for the other uncertain parameters such as the incubation time, the time until an exposed individual becomes infectious, the probability of ending up in the ICU, and the fraction of symptomatic patients that goes into self-quarantine. Of course, more scenarios can be created by varying some of these uncertain parameters as well, but this will only yield more alternative scenarios that offer a possible explanation of the observed death toll and ICU admissions.

In case further research resolves one or more of the uncertainties in virus and/or disease characteristics, then our or a similar study can be used to narrow down the range of possibilities for the other characteristics. Additionally, a better estimate of the death toll allows for further reducing the number of realistic scenarios. For example, if the lower bound on the death toll estimate were increased by 50% and the upper bound reduced by 33%, only 10 scenarios remain. The bandwidth of the predictions on infections and immunity is already much smaller than for the original 18 scenarios: the percentage of the population that was infected varied from 1.8% to 3%, and the level of immunity ranged from 1.3% to 2.8%.

The values for virus transmission probability reported in this study should be interpreted with caution. First, there is uncertainty whether the transmission probability is constant when one is infected, or if the probability changes over time depending on the disease stage of an individual. This is an example of one of the uncertainties that has not been included in the study and including it will only enlarge the number of plausible scenarios. Second, the daily number of infected individuals in our simulation was determined by multiplying this virus transmission probability with the number of infectious individuals and their number of daily contacts. Here, the definition of a contact is important: including only conversations as contacts results in fewer social contacts than including also e.g. passersby. A restricted definition of contacts naturally corresponds to a higher transmission probability. This was reflected by the model as well: a more restricted definition leads to fewer contacts, hence a higher transmission probability was necessary to achieve the same number of infections, ICU admissions and deaths. We therefore did not consider our validated values for the virus transmission probabilities to be exact and universally applicable. Rather we showed that for a variety of assumptions on CFR, developing immunity and symptom development, and for a given definition of contacts, there exist transmission probabilities that lead to realistic simulation results. Further research is required to have a concrete definition of a contact in combination with the probability of transmitting the virus.

In summary, this paper presents a comprehensive study to test the validity of a wide range of SARS-CoV-2 virus- and COVID-19 disease characteristics. A variety of assumptions yielded a realistic number of deaths and daily ICU admissions. However, these scenarios disagreed on the predicted number of infections and immune individuals, two unobservable but important

metrics. From this we conclude that the currently available information on the behavior of the SARS-CoV-2 virus is insufficient to accurately model and predict the virus spread, evaluate the effects of social distancing measures in detail and develop social distancing policies or even exit strategies. Note that this does not imply that such studies are not informative: epidemiological forecasting models have helped us understand the severity of the pandemic already early on, and are well capable of forecasting the trend of the virus spread. When conducting a forecasting analysis that requires insights in the infections and immunity among a population we highly recommend to analyze the results for several scenarios on virus- and disease characteristics, report the range of results obtained with these scenarios and draw conclusions and recommendations based on the full set of scenario-specific outcomes.

## Supporting information

**S1 File.**
(ZIP)

**S1 Data. The data required to duplicate the research.** It contains the data files in the input folder and the source code of the model in the src folder. Furthermore, in order to understand the input data please read the Readme file. Finally, in case you are unable to work with java, there is a .jar file included that runs our code without having to understand java.
(ZIP)

## Acknowledgments

We thank Dr. Jean-Luc Murk of the Elisabeth-Tweesteden hospital Tilburg, the Netherlands, for his valuable input and feedback.

## Author Contributions

**Conceptualization:** Marleen Balvert, Hein Fleuren.

**Methodology:** Melissa Koenen, Marleen Balvert, Ruud Brekelmans, Valentijn Stienen, Joris Wagenaar.

**Software:** Melissa Koenen.

**Supervision:** Hein Fleuren.

**Validation:** Valentijn Stienen, Joris Wagenaar.

**Visualization:** Valentijn Stienen.

**Writing – original draft:** Marleen Balvert, Joris Wagenaar.

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
