## [Decision Letter · Decision Letter 0]

12 Nov 2020

PONE-D-20-29479

Forecasting the spread of SARS-CoV-2 is inherently ambiguous given the current state of virus research

PLOS ONE

Dear Dr. Wagenaar,

Thank you very much for submitting your manuscript "Forecasting the spread of SARS-CoV-2 is inherently ambiguous given the current state of virus research" (PONE-D-20-29479) for consideration at PLOS ONE. As with all papers reviewed by the journal, your manuscript was reviewed by members of the editorial board and by several independent reviewers. In light of the reviews (below this email), we would like to invite the resubmission of a significantly-revised version that takes into account the reviewers' comments.

We look forward to receiving your revised manuscript.

Kind regards,

Abdallah M. Samy, PhD

Academic Editor

PLOS ONE

**Journal Requirements:**

**Reviewers' comments:**

Reviewer's Responses to Questions

**Comments to the Author**

1. Is the manuscript technically sound, and do the data support the conclusions?

Reviewer #1: Partly

Reviewer #2: Yes

2. Has the statistical analysis been performed appropriately and rigorously? 

Reviewer #1: No

Reviewer #2: Yes

3. Have the authors made all data underlying the findings in their manuscript fully available?

Reviewer #1: Yes

Reviewer #2: Yes

4. Is the manuscript presented in an intelligible fashion and written in standard English?

Reviewer #1: Yes

Reviewer #2: Yes

5. Review Comments to the Author

Reviewer #1: This article entitled Forecasting the spread of SARS-CoV-2 is inherently ambiguous given the current state

of virus research presents a statistical model adapted from standard epidemiological models.

Its objective is to show us that existing models should be viewed with caution given the lack of knowledge on the viral characteristics of SARS-CoV-2.

The authors therefore offer us a new model, necessarily based on assumptions (not the same as the previous models) and vary certain parameters of interest to explain to you at the end that various variations on distinct parameters can lead to a fairly high prediction. This is the very principle of probabilities and statistics: an accumulation of errors can nevertheless lead to a fair result. And because of this, it is very difficult to say that one model is more or less fair than another, simply because the data and parameters are often inherent in the structure that creates the model.

In the case of this article there is nothing new about varying parameters and checking the impact they have on the dynamics of SARS-CoV-2. It is very judicious to want to take into account the movements of population according to time but where are the elementary parameters accounting for the functioning of the virus, for example it is completely false to suppose that the contagiousness of an individual of a class of age is strictly that of another individual of the same age group at time t. The contagion of each individual very probably evolves according to his viral load and the viral load is not constant over time.

This article entitled Forecasting the spread of SARS-CoV-2 is inherently ambiguous given the current state

of virus research

presents a statistical model adapted from standard epidemiological models. Its objective is to show us that existing models should be viewed with caution given the lack of knowledge on the viral characteristics of SARS-CoV-2. The authors therefore offer us a new model, necessarily based on assumptions (not the same as the previous models) and vary certain parameters of interest to explain to you at the end that various variations on distinct parameters can lead to a fairly high prediction. fair. This is the very principle of probabilities and statistics: an accumulation of errors can nevertheless lead to a fair result. And because of this, it is very difficult to afford to say that one model is more or less fair than another, simply because the data and parameters are often inherent in the structure that creates the model.

In the case of this article one thing is for sure, there is nothing new about varying parameters and checking the impact they have on the dynamics of SARS-CoV-2. It is very judicious to want to take into account the movements of population according to time but where are the elementary parameters accounting for the functioning of the virus, for example it is completely false to suppose that the contagiousness of an individual of a class of age is strictly that of another individual of the same age group at time t. The contagion of each individual very probably evolves according to his viral load and the viral load is not constant over time.

Likewise, to make the model more complex, it would be wise to take into account an infection reduction coefficient which is due to the implementation of health measures (for example homeworking).

Finally, it seems to me very difficult conceptually to vary certain parameters but to fix others while explaining to us that it is very dangerous to set parameters to evaluate the dynamics of a virus. From this perspective, it would have been undoubtedly much more interesting and coherent to try to present a nonparametric model of evolution of the dynamics of SARS-CoV-2.

Reviewer #2: In the present paper titled “Forecasting the spread of SARS-CoV-2 is inherently ambiguous given the current state of virus research”, the authors addressed the validity of various assumptions using an epidemiological simulation model. Result feedback that multiple scenarios all lead to

realistic numbers of deaths and ICU admissions, two observable and verifiable metrics,

but gave different estimates for the number of infected and immune individuals. To validate the assumption on the spread of virus or disease, the present paper applied a popular classical model called the SEIR model and agent-based simulation which can address the challenges in the SEIR model.

The study was timing and output were interesting and shows its originality. The paper was well structured, the method and materials for assumption and the corresponding results, technical support is sound enough. However, the authors may wish to consider minor revisions as follows to the manuscript:

• The reader may benefit from a definition of the SEIR model and agent-based simulation with short theory.

• In the abstract review, more considerable information should be given to represent the whole contributions of the present manuscript.

• If possible it is suggested to add legend on figures.

6. PLOS authors have the option to publish the peer review history of their article (what does this mean?). If published, this will include your full peer review and any attached files.

Reviewer #1: No

Reviewer #2: No

---

## [Author Response · Author response to Decision Letter 0]

11 Dec 2020

We would like to thank the Associate Editor and the Referees for the careful consideration of our work and for giving us the opportunity to address the comments on the earlier version of our manuscript. We outline our response to the comments below. We hope our responses address the concerns of both Reviewer 1 and 2 and we remain at the disposal of the referee team for further clarifications.

Before we respond to the comments by Associate Editor and referees, we would like to remark that while preparing the revision we noted an error in the input data used for our simulations. Accidentally, the wrong daily contact patterns were used for one of the age groups. We ran the simulations again with the correct contact patterns and adapted the results in the tables and text of the revised manuscript. The conclusions remain valid: multiple scenarios lead to realistic numbers of verifiable metrics (number of deaths and ICU admissions) but result in varying results for the number of infections and immune individuals.

Response to Associate Editor’s Comments

Comment: Please include the following items when submitting your revised manuscript: 

· A rebuttal letter that responds to each point raised by the academic editor and reviewer(s). You should upload this letter as a separate file labeled 'Response to Reviewers'. 

· A marked-up copy of your manuscript that highlights changes made to the original version. You should upload this as a separate file labeled 'Revised Manuscript with Track Changes'. 

· An unmarked version of your revised paper without tracked changes. You should upload this as a separate file labeled 'Manuscript'. 

Response: We have included both a rebuttal letter where we address all points raised by the editor and by the reviewers. Further, we have included a marked-up copy of our manuscript that highlights all changes we made to the original version. This copy crosses all parts of the original text that we do no longer use and inserts all new text in red. Further, we have also included our new version without the tracked changes.

Response to reviewer 1’s comments

Comment: This article entitled “Forecasting the spread of SARS-CoV-2 is inherently ambiguous given the current state of virus research” presents a statistical model adapted from standard epidemiological models. 

Its objective is to show us that existing models should be viewed with caution given the lack of knowledge on the viral characteristics of SARS-CoV-2. 

 The authors therefore offer us a new model, necessarily based on assumptions (not the same as the previous models) and vary certain parameters of interest to explain to you at the end that various variations on distinct parameters can lead to a fairly high prediction. This is the very principle of probabilities and statistics: an accumulation of errors can nevertheless lead to a fair result. And because of this, it is very difficult to say that one model is more or less fair than another, simply because the data and parameters are often inherent in the structure that creates the model. 

 In the case of this article there is nothing new about varying parameters and checking the impact they have on the dynamics of SARS-CoV-2. It is very judicious to want to take into account the movements of population according to time but where are the elementary parameters accounting for the functioning of the virus, for example it is completely false to suppose that the contagiousness of an individual of a class of age is strictly that of another individual of the same age group at time t. The contagion of each individual very probably evolves according to his viral load and the viral load is not constant over time.

Response: Thank you for your comment and insights. We agree that models inherently have statistical properties. Our goal was to provide insight in which combinations of scenarios/parameters seem plausible, which in the end turned out to be notoriously difficult and it leads to widely varying results on currently unknown outcomes (e.g. immunity and number of infected people). Statistics inherently leads to variations, however, these variations seem to be even stronger with SARS-CoV-2 due to the lack of knowledge and the difficulty to track down this lack. This seems to have been forgotten by published simulation studies in the past months.

 In case we include even more uncertain parameters, as for example the development of the contagiousness of an infected individual, the number of plausible parameters/scenarios would most likely increase even more. We have tried to explain this more extensively in the new version of the paper in the Introduction section on lines 30-34 on page 2, and in the Conclusion section on lines 416-426 on page 13-14, with the paragraph:

Epidemiological models such as ours are based on many input parameters, most of which are uncertain and only have crude estimates. In our simulation study, we already considered many scenarios by varying only four input parameters: the probability of developing symptoms, the case fatality ratio, the probability of developing immunity, and the virus transmission probability. The best available estimates from literature were used for the other uncertain parameters such as the incubation time, the time until an exposed individual becomes infectious, the probability of ending up in the ICU, and the fraction of symptomatic patients that goes into self-quarantine. Of course, more scenarios can be created by varying some of these uncertain parameters as well, but this will only yield more alternative scenarios that offer a possible explanation of the observed death toll and ICU admissions.

 Further, our simulation does not include the infection probability at the individual level, but at an aggregate level where group characteristics are taken into account. As such, the simulation is not able to include contagiousness levels increasing over time. However, it would not change the average outcome of our simulation, because in the end an individual will on average have the same contagiousness level as we included. We have tried to explain this better in the second and third paragraph in the Methods and Materials section on page 3 (lines 68-100).

Comment: Likewise, to make the model more complex, it would be wise to take into account an infection reduction coefficient which is due to the implementation of health measures (for example homeworking).

Response: Thank you for your comment and suggestion of including infection reduction coefficients. First, it would be possible to include a reduction coefficient, but this would only lead to more uncertainty as it is unknown which fraction of the people followed the health measures. Even if the reduction coefficient is known, then there is uncertainty as to how it relates to the total contact reduction of the inhabitants. Including this uncertainty would give us more possible parameter combinations that might be correct. We have included in the new version of the paper in the Introduction on lines 30-34 on page 2 and Conclusion section on lines 416-426 the statement that we already take four levels of uncertainty into account, and including more would only lead to having even more plausible parameter combinations. 

Second, it is not the objective of our paper to look at the effects of health measures. More fundamentally, we would like to model the virus behavior as purely as possible in order to apply it later in other settings (e.g. humanitarian related settings as refugee camps and slums). In order to do so, more research is first required in finding the correct parameters. See the following added paragraph in the new version of the paper in the Introduction section on lines 15-21:

The aim of this paper is to get a good view on the spread of the virus and its characteristics as the virus spread would behave without taking any social distancing measures. Background is that, in future research, we are interested how the SARS-CoV-2 virus spreads in low income countries, slums and refugee camps where, for various reasons, measures hardly can be taken or are not effective at all. We therefore use data of the initial phase of the COVID-19 spread in the Netherlands where relatively plenty of good quality data is available.

Comment: Finally, it seems to me very difficult conceptually to vary certain parameters but to fix others while explaining to us that it is very dangerous to set parameters to evaluate the dynamics of a virus. From this perspective, it would have been undoubtedly much more interesting and coherent to try to present a nonparametric model of evolution of the dynamics of SARS-CoV-2.

Response: In general, most epidemiological models assume fixed parameters and then evaluate the spread of a virus. Our paper demonstrates that varying four disease and virus characteristics already leads to a wide variety of results, which can all be correct given the current knowledge of the virus. The best available estimates from literature were used for the other uncertain parameters such as the incubation time, the time until an exposed individual becomes infectious, the probability of ending up in the ICU, and the fraction of symptomatic patients that goes into self-quarantine.

The four parameters for which we vary the values are all very uncertain in literature. In case we would vary the other parameters as well, then that would lead to more variety and thus more possible correct parameters. We have tried to explain this more carefully in the new version of the paper in the Introduction on lines 30-34 and in the Conclusion section on lines 416-426.

 Furthermore, non-parametric models need much data and do not make use of the structure of the underlying model. The structure of our problem is known and should therefore be used, only the parameters within the model are uncertain. Therefore, we have decided to use the simulation model as presented in the paper. We have tried to explain this reasoning in the first three paragraphs of the Methods and Materials section (lines 58-100) in the new version of the paper.

Response to reviewer 2’s comments

Comment: In the present paper titled “Forecasting the spread of SARS-CoV-2 is inherently ambiguous given the current state of virus research”, the authors addressed the validity of various assumptions using an epidemiological simulation model. Result feedback that multiple scenarios all lead to realistic numbers of deaths and ICU admissions, two observable and verifiable metrics, but gave different estimates for the number of infected and immune individuals. To validate the assumption on the spread of virus or disease, the present paper applied a popular classical model called the SEIR model and agent-based simulation which can address the challenges in the SEIR model. 

 The study was timing and output were interesting and shows its originality. The paper was well structured, the method and materials for assumption and the corresponding results, technical support is sound enough. However, the authors may wish to consider minor revisions as follows to the manuscript: 

 • The reader may benefit from a definition of the SEIR model and agent-based simulation with short theory. 

 • In the abstract review, more considerable information should be given to represent the whole contributions of the present manuscript. 

 • If possible it is suggested to add legend on figures. 

Response: Thank you for the feedback and the suggestions to improve the paper. We have added a clear definition of a SEIR model and agent-based simulations in the first three paragraphs of the Methods and Materials section on page 3 (lines 58-100) in the new version of the paper:

In the existing literature, many researchers have already pointed out that classical SEIR models and agent-based simulations need to be adapted in order to catch all the important virus characteristics of COVID-19. Although SEIR models have been commonly used to model disease spread and form the basis of many of today’s COVID-19 epidemiological models [11,12], they need to be adapted to differentiate between age groups or geographic locations [1,13-16]. An often used alternative is agent-based simulation [3,17,18], which allows for modeling at the individual level rather than aggregating over the entire population. This is important when modeling COVID-19 [19,20], as it allows for social and travel patterns that depend on age group and location.

The simulation model that we use to validate assumptions on virus spread and disease progression holds the middle between a compartmented SEIR model and an agent-based simulation model. Traditional compartmented SEIR models divide the population into several health stages such as susceptible (healthy individuals, denoted as S, exposed (asymptomatic infected individuals who are not able to spread the virus, E), infected (symptomatic infected individuals, I) and recovered (immune or deceased individuals, R). Over time, the health condition of individuals may progress from one health stage to another. Within the population no distinction is made between individuals based on age or other personal characteristics. Virus parameters are often estimated using differential equations. While the spread of SARS-CoV-2 as well as the disease progression behave differently depending on an individual's age and location (rural or urban), splitting the population into subgroups based on these characteristics complicates the derivation of model parameters and increases the risk of nonidentifiability. In agent-based simulation one simulates the exact daily movements of each individual. As a result agent-based simulation can take many (virus-related) individual characteristics into account and can simulate the daily contacts of each individual in detail.} 

While agent-based models provide the level of detail required to model SARS-CoV-2 that is lacking in SEIR models, the computational complexity of agent-based modeling is prohibitive for a population of 17 million people, which is the size of the population of the Netherlands. We therefore propose an intermediate modeling form: a coarse-grained agent-based simulation model which uses the idea of health stages to simulate agents, where agents are characterized only by their age group and geographic region. The model is akin to agent-based simulations [3,17,18] in that distinctive individuals are simulated who commute between their region of residence and region of employment. The main difference is that we do not include social interactions at an individual level but aggregate over groups of people with the same age, region of residence and work region. This allows us to simulate on a large scale, i.e. to simulate all inhabitants of a country or state, while including individual characteristics such as age, region of residence and commute patterns that are highly relevant when modeling the spread of SARS-CoV-2 [7,21]. Compared to an agent-based simulation our model reduces the number of assumptions one needs to make, as we aim to simulate on a country level. Furthermore, it gives us more freedom than when using the compartment model.

Furthermore, we have changed the abstract such that it contains more information about the contributions of our manuscript and we have added legends to the figures and hope this helps to explain the figures in a better way.

---

## [Decision Letter · Decision Letter 1]

2 Jan 2021

Forecasting the spread of SARS-CoV-2 is inherently ambiguous given the current state of virus research

PONE-D-20-29479R1

Dear Dr. Wagenaar,

We’re pleased to inform you that your manuscript, "Forecasting the spread of SARS-CoV-2 is inherently ambiguous given the current state of virus research" (PONE-D-20-29479R1), has been judged scientifically suitable for publication and will be formally accepted for publication once it meets all outstanding technical requirements.

Kind regards,

Abdallah M. Samy, PhD

Academic Editor

PLOS ONE

---

## [Editor Report · Acceptance letter]

11 Feb 2021

PONE-D-20-29479R1 

Forecasting the spread of SARS-CoV-2 is inherently ambiguous given the current state of virus research 

Dear Dr. Wagenaar:

I'm pleased to inform you that your manuscript has been deemed suitable for publication in PLOS ONE. Congratulations! Your manuscript is now with our production department. 

Kind regards, 

on behalf of

Dr. Abdallah M. Samy 

Academic Editor

PLOS ONE